# Substantiation and Effectiveness of Remote Monitoring System Based on IoMT Using Portable ECG Device

**DOI:** 10.3390/bioengineering11080836

**Published:** 2024-08-16

**Authors:** Hee-Young Lee, Yoon-Ji Kim, Kang-Hyun Lee, Jung-Hun Lee, Sung-Pil Cho, Junghwan Park, Il-Hwan Park, Hyun Youk

**Affiliations:** 1Digital Health Laboratory, Yonsei University Wonju College of Medicine, Wonju 26417, Gangwon State, Republic of Korea; hylee3971@yonsei.ac.kr (H.-Y.L.); rladbswl25@yonsei.ac.kr (Y.-J.K.); 2Department of Emergency Medicine, Yonsei University Wonju College of Medicine, Wonju 26426, Gangwon State, Republic of Korea; ed119@yonsei.ac.kr (K.-H.L.);; 3MEZOO Co., Ltd., Wonju 26354, Gangwon State, Republic of Korea; spcho@me-zoo.com (S.-P.C.); jhpark@me-zoo.com (J.P.); 4Regional Trauma Center, Wonju Severance Christian Hospital, Wonju 26426, Gangwon State, Republic of Korea; nicecs@yonsei.ac.kr

**Keywords:** remote monitoring system, internet of medical things (IoMT), portable ECG device, cardiovascular disease

## Abstract

Cardiovascular disease is a major global health concern, with early detection being critical. This study assesses the effectiveness of a portable ECG device, based on Internet of Medical Things (IoMT) technology, for remote cardiovascular monitoring during daily activities. We conducted a clinical trial involving 2000 participants who wore the HiCardi device while engaging in hiking activities. The device monitored their ECG, heart rate, respiration, and body temperature in real-time. If an abnormal signal was detected while a physician was remotely monitoring the ECG at the IoMT monitoring center, he notified the clinical research coordinator (CRC) at the empirical research site, and the CRC advised the participant to visit a hospital. Follow-up calls were made to determine compliance and outcomes. Of the 2000 participants, 318 showed abnormal signals, and 182 were advised to visit a hospital. The follow-up revealed that 139 (76.37%) responded, and 30 (21.58% of those who responded) sought further medical examination. Most visits (80.00%) occurred within one month. Diagnostic approaches included ECG (56.67%), ECG and ultrasound (20.00%), ultrasound alone (16.67%), ECG and X-ray (3.33%), and general treatment (3.33%). Seven participants (23.33% of those who visited) were diagnosed with cardiovascular disease, including conditions such as arrhythmia, atrial fibrillation, and stent requirements. The portable ECG device using the patch-type electrocardiograph detected abnormal cardiovascular signals, leading to timely diagnoses and interventions, demonstrating its potential for broad applications in preventative healthcare.

## 1. Introduction

In South Korea, remote monitoring in healthcare encounters significant legal obstacles due to the Medical Service Act, particularly Article 34, which prohibits remote diagnosis and treatment between doctors and patients [1]. This article mandates that medical services must be delivered face-to-face, emphasizing the necessity of in-person interactions for diagnosis and treatment. Although the first telemedicine pilot project was attempted in 1988, the debate over its introduction continues today. On the one hand, promoting and activating the telemedicine business along with the development of information and communication technology is an important task for the country in terms of industrial entry, and urges the promotion of the project, saying that science and technology will benefit the public by providing quality medical services. On the other hand, they say that medical practice that does not require face-to-face contact is still risky and that there is no need to pursue this project while taking such risks [2]. As a result, the law restricts the growth of telemedicine and remote monitoring technologies within the country.

Conversely, countries like the United States and those in the European Union have adopted more liberal legal frameworks and economic incentives, which facilitate the rapid development and widespread integration of remote monitoring technologies into their healthcare systems. These nations have embraced telemedicine and remote monitoring, recognizing their potential to enhance healthcare access, improve patient outcomes, and reduce costs. In the United States, remote monitoring has been extensively utilized to manage various health conditions, including COVID-19, chronic obstructive pulmonary disease (COPD), and heart failure. In Japan, remote monitoring is also being applied in diverse healthcare settings, including neonatal care and immunization safety monitoring. In the United States, the Providence health system implemented a Home Monitoring Program (HMP) for COVID-19 patients in March 2020. This program utilized at-home pulse oximeters, thermometers, and text-based surveys to monitor symptoms. The HMP was found to be highly effective, with over 80% engagement in daily text-based surveys and high levels of comfort using home monitoring devices among both English- and Spanish-speaking participants. Additionally, the program was associated with increased outpatient and emergency department encounters, indicating that patients felt safer and more satisfied monitoring their condition from home [3]. Similarly, remote patient monitoring for chronic diseases such as COPD has shown potential benefits, including the early detection of exacerbations, prompt access to therapy, and ultimately improved patient outcomes [4]. In heart failure management, telemonitoring strategies became routine during the pandemic, with evidence supporting the efficacy and safety of virtual visits and remote vital sign monitoring. The intensity of monitoring was tailored to match the patient’s risk profile, highlighting the importance of a structured approach to telehealth implementation [5]. In Japan, remote monitoring is utilized in various healthcare contexts, including the monitoring of adverse events following immunization (AEFI). The Japanese AEFI monitoring system ensures the safe implementation of immunizations by tracking and addressing adverse events. This system is complemented by a unique compensation mechanism, known as the relief system, which provides support for individuals affected by immunization-related issues [6]. Additionally, remote monitoring is employed in neonatal intensive care units (NICUs) in Japan. The use of transcutaneous (tc) measurements of partial pressure of oxygen (tcPO2) and carbon dioxide (tcPCO2) in NICUs allows for the continuous and non-invasive monitoring of critically ill neonates, ensuring timely interventions and better health outcomes [7]. As such, Korea has a problem that reduces market dynamism due to positive regulations based on laws that list only those things that are permitted, thereby hindering the growth of the industry along with the flow of the Fourth Industrial Revolution [8].

As a measure to solve this problem, the Ministry of SMEs and Startups of Korea has established a system to designate the Regulation-Free Special Zone (RFZ) by region and to ease key regulations related to new businesses as a package, in order to create an environment that can focus on rapidly changing technological industrialization in the era of the Fourth Industrial Revolution [9]. First of all, a study analyzing the legal and institutional issues of regulations recognized as obstacles in industrial sites was reported [10,11,12], and in 2019, the Gangwon Digital Healthcare Regulation-Free Special Zone was designated to make it grow the digital healthcare industry, making it possible for Gangwon State to conduct empirical research as a test bed for regulatory improvement of various laws, as shown in Table 1 below.

In this study, in order to ease the regulations under Article 34 of the Medical Act of Korea, a biosignal monitoring healthcare service was planned that partially applied that wearable devices for biosignal monitoring would be provided and that a doctor in the hospital would be able to monitor electrocardiograms (ECGs) for rapid treatment and rescue in emergency situations. Then, an empirical study was conducted on remote monitoring between a doctor and patients depending on the limited scope, location, and method. The ECG is a primary diagnostic tool for various cardiovascular diseases, including arrhythmias, myocardial infarctions, and other cardiac abnormalities. The high diagnostic accuracy of ECGs in detecting myocardial ischemia was demonstrated, reinforcing the critical role of ECGs in early detection and treatment planning for cardiovascular conditions [13]. Also, studies such as those by Mincholé and Rodríguez have highlighted the predictive power of ECG signals in identifying patients at risk of future cardiac events, further underscoring the importance of ECG-based diagnostics [14]. In addition, the remote monitoring of ECG signals has become increasingly important, especially with the rise of chronic heart disease and the need for continuous, real-time monitoring. A report by the American Heart Association (AHA) in 2019 emphasized that remote monitoring can significantly improve patient outcomes by enabling the early detection of potential issues, thereby allowing for timely medical interventions [15]. It was found that the remote monitoring of heart failure patients reduced hospitalizations and improved survival rates. These findings highlight the critical role of remote ECG monitoring in managing cardiovascular health, especially in at-risk populations [16]. 

However, despite the various studies reported on ECG signal monitoring, some shortcomings may affect its efficacy and reliability. These shortcomings can be broadly categorized into device power, signal quality, and practical use. One of the primary disadvantages is related to the power constraints of ECG monitoring devices, particularly implantable ones. The limited battery capacity restricts the functionalities of these devices, preventing them from achieving their full potential. Although wireless power transfer (WPT) technology has been proposed to address this issue, the implementation and efficiency of such systems in real-world scenarios remain to be thoroughly validated [17]. Signal quality is another significant challenge in ECG monitoring. During data acquisition and transmission, various artifacts can degrade the ECG signal, leading to incorrect clinical diagnoses. These artifacts may stem from environmental noise, patient movement, or device-related issues. While advanced signal processing techniques such as wavelet transforms, neural networks, and adaptive filters have been developed to suppress these artifacts, ensuring consistently high-quality signals remains a complex task. The proposed multistage adaptive filter aims to tackle multiple types of artifacts, but it requires prior knowledge of the interferences, which may not always be feasible [18]. Wearable ECG devices, such as patch-type monitors, introduce their own set of challenges. These devices, while unobtrusive and user-friendly, are typically suited for short- to medium-term monitoring (days to weeks) rather than long-term continuous use. Moreover, achieving a balance between low power consumption and high diagnostic accuracy in these devices remains a significant challenge. Existing schemes often suffer from high power consumption to maintain accuracy, which can limit their practicality [19]. Another critical issue is the difficulty in ensuring the clinical acceptability of ECG signals in unsupervised environments. Real-time signal quality assessment (SQA) methods have been proposed to classify the acquired ECG signal into acceptable or unacceptable categories, but these methods need to be robust and efficient to be effective in practical applications [20].

The Internet of Medical Things (IoMT) plays a crucial role in modern healthcare by facilitating remote monitoring. It enables the collection and analysis of patients’ vital signs and health data in real time, allowing healthcare professionals to continuously observe patients’ conditions. This improves the efficiency of patient management and allows for quick responses in emergencies, ultimately enhancing patient safety and treatment outcomes [21]. IoMT is a term for medical objects that apply IoT technology to the medical field and can transmit medical information and biosignal data in real time through network communication. IoMT devices include an integrated circuit for data collection, a signal processing device for data processing, and a network module through data transmission. So, this system can comprise a minimal human intervention and effectively monitor things that were not monitored before. Digital health, particularly involving IoMT devices, is revolutionizing health management in daily life. Lupton discussed how IoMT devices, such as wearable ECG monitors, provide continuous health data that can be analyzed to offer personalized health insights and interventions [22]. The advantages of using IoMT devices include improved patient adherence to treatment plans, real-time health monitoring, and the ability to collect large datasets for advanced analytics, which can lead to more accurate predictive models and personalized healthcare strategies. Also, it was demonstrated that digital health technologies not only improve health outcomes but also enhance the quality of life by allowing individuals to manage their health proactively and conveniently [23].

This study aimed to verify the effectiveness of remote biological signal monitoring technology using an IoMT device.

## 2. Materials and Methods

### 2.1. Participants

In this study, a clinical trial was conducted to verify the effectiveness of remote monitoring diagnosis technology using an IoMT device. The clinical trial was conducted a total of 30 times in about 8 months and a total of 2000 hikers participated. The inclusion criteria were adults aged 19 years or older, who gave written consent to participate in the empirical research, and who could understand and follow the on-site training and instructions. Also, the exclusion criteria encompassed those who could not attach the patch-type IoMT device due to skin disease, those who were pregnant, those who were participating in another clinical trial at the same time, or those who had recently completed participation in a previous clinical trial. Each hiker filled out a survey about the clinical trial consent and existing health information in advance, then attached a wearable ECG monitoring patch to their chest and climbed a mountain. This study was carried out with approval from the Research Ethics Committee of Yonsei University Wonju Severance Christian Hospital (IRB approval number: CR319186) and registered with the Clinical Research Information Service (CRIS) operated by the Korea Disease Control and Prevention Agency (CRIS trial registration number: KCT0005795).

### 2.2. ECG Monitoring Device

HiCardi^®^ (MEZOO Co. Ltd., Wonju, Gangwon State, Republic of Korea), an IoMT device used in this study, is a patch-type electrocardiograph. This wearable device could monitor and record various parameters related to heart signals such as a single-lead ECG, heart rate, respiration, skin surface temperature, and activity. It has a comprehensive set of specifications aimed at providing accurate health monitoring data. The heart rate (HR) range of the device is from 0 to 300 beats per minute (bpm), with a precision of ±2 bpm or ±2%, whichever is greater. The temperature monitoring range is from 32 °C to 43 °C, with an accuracy of ±0.3%. It also provides respiration rate measurements ranging from 5 to 60 breaths per minute or up to 120 breaths per minute, maintaining an accuracy of ±2 bpm or ±2%. The device offers an impressive battery operation time of 72 to 168 h with a charging time of 1 to 2 h. It also features water protection capabilities from IPX2 to IP67, ensuring durability and reliability in various conditions. The memory capacity of the device supports data recording for up to 7 days, with optional expansions available. Despite its robust functionality, the device maintains a lightweight design, weighing less than 10 g (0.35 oz). The data from the wearable patch were transferred through Bluetooth low energy to a mobile gateway, which was implemented as a smartphone application. The mobile gateway transmitted the real-time data to the core, analytics, and service platform [24].

### 2.3. Role of the Physician at the IoMT Monitoring Center

Wonju Severance Christian Hospital operates an IoMT monitoring center for digital health research, including remote monitoring and non-face-to-face collaboration. At the IoMT monitoring center, a physician, who was an emergency medicine specialist, collected health status information such as basic information, vital signs (blood pressure and body temperature), medical history, and medications surveyed by participants at the empirical research site. After that, he monitored the hikers’ biosignals transmitted in real time to check whether an abnormal ECG was observed. If an abnormal signal was detected during ECG biosignal monitoring and disease was suspected, the physician notified the clinical research coordinators (CRCs) at the empirical research site, and then the CRCs advised the participant to visit a hospital. Also, if problems such as noise in the device or disconnection due to poor attachment occurred, the problem was resolved through on-site response. In the event of an emergency, the condition of the participant was checked according to the emergency response protocol, and the patient’s condition and location information was delivered to the National Fire Agency 119 to prepare for rapid emergency treatment through emergency dispatch (Figure 1).

### 2.4. Empirical Research Scenarios

Figure 2 shows the flowchart of IoMT monitoring service. Before starting the clinical trial, the CRCs explained the purpose of the clinical trial. Those who agreed to participate in the clinical trial completed a survey on their physical information and health status. When selected as participants, they were provided with an IoMT medical device, installed an application linked to the device, and were instructed on how to use it and the precautions. Participants attached the IoMT medical device linked to the application to a designated location and began measuring biometric information. The measured biometric information was transmitted to the IoMT monitoring center in real time. An emergency medical specialist remotely monitored the participants’ vital signs at the center. Participants wore the IoMT medical device and took the clinical test while hiking. However, in the event of an emergency, the test subject’s emergency situation was responded to by contacting the National Fire Agency (119) according to established protocols. Once monitoring of the IoMT medical device was completed, biosignal changes were observed and tracked by exercise load. The emergency medicine specialist identified the presence or absence of any adverse reactions that occurred while hiking, and the participants completed the clinical trial by returning the IoMT medical devices after completing a satisfaction survey regarding their health status and service model.

### 2.5. Follow-Up Management of Participants Who Were Recommended to Visit a Hospital

Follow-up observations were conducted through telephone surveys on participants who were recommended to receive hospital treatment after abnormal signals were discovered at the demonstration site. A survey on telephone calls was conducted regarding whether or not the patient visited the hospital, the date of the hospital visit, the method of diagnosis, and the results of the diagnosis or treatment.

### 2.6. Satisfaction Survey

At the end of the empirical research, participants completed a satisfaction survey to evaluate their experiences. The survey included questions to assess various aspects of their participation and the usefulness of the provided health information and devices. Participants were asked about their previous experiences in other clinical studies and whether the health information provided (ECG, heart rate, respiration, body temperature) was helpful. They were also asked if they found the IoMT device to be helpful for health management. Satisfaction with the IoMT device was measured using a five-point Likert scale ranging from “Very Satisfied” to “Very Dissatisfied”. Specific aspects evaluated included discomfort experienced after attaching the device (e.g., detachment, itchiness, restriction of movement), trust in the provided health information, and willingness to use the attached IoMT device in the future. Additionally, satisfaction with the HiCardi mobile application was assessed, particularly the ease of accessing health information through the application, using a five-point Likert scale. Participants also rated their overall satisfaction with the study and were invited to provide comments on potential improvements. The responses were collected and analyzed to gauge overall satisfaction and identify areas for improvement. Feedback from participants provided valuable insights into the user experience and the effectiveness of the health information and devices used in the study.

## 3. Results

### 3.1. General Characteristics

Table 2 presents the demographic characteristics of the participants. A total of 2000 people participated, including 811 men and 1189 women. The average age of the participants was 49.36 ± 14.65 years old. Also, the males had a mean age of 49.35 years, while the females had a mean age of 49.37 years. The age distribution shows that the highest proportion of hikers were in their 50s (28.05%), followed by those in their 60s (21.40%) and 40s (17.65%). Among the men, the 50s (23.18%) and 60s (23.05%) age groups were more prominent, whereas a higher proportion of the women were in their 50s (31.37%) and 60s (20.26%). Notably, 4.65% of the hikers were in their 70s or older, with a higher proportion of men (7.15%) compared to women (2.94%). In terms of the physical characteristics, there are clear gender differences. The average height of all the hikers was 164.16 cm and the average weight was 64.26 kg. The men have an average height of 171.35 cm and an average weight of 73.04 kg, while the women average 159.26 cm in height and 58.26 kg in weight. The blood pressure measurements showed a mean systolic blood pressure (SBP) of 130.89 mmHg and a mean diastolic blood pressure (DBP) of 85.61 mmHg. The men had a higher average SBP (136.85 mmHg) and DBP (88.79 mmHg) compared to the women, who have average values of 126.82 mmHg and 83.44 mmHg, respectively. The mean pulse rate (PR) was 74.87 bpm, with only slight differences between the men (76.05 bpm) and women (74.07 bpm). The average body temperature is 36.72 °C. Regarding health status, 66.25% of the hikers reported having no chronic diseases, with 63.87% of men and 67.87% of women reporting the same. The prevalence of chronic diseases among hikers includes one disease in 21.10%, two diseases in 9.35%, and more than three diseases in 3.30%. For the family history, 45.30% of the hikers reported having a family history of health issues, with a higher proportion among women (51.22%) compared to men (36.62%).

### 3.2. Detection of Abnormal Signals and Recommendation to a Hospital through Participant ECG Monitoring

Table 3 summarizes the results of the clinical trials, focusing on the number of participants, the detection of abnormal ECG signals, recommendations to visit the hospital, and detailed cardiovascular disease findings. A total of 2000 participants were involved across 30 trials. Among these participants, 318 (15.90%) were monitored for abnormal signals on their ECGs. Also, 296 (93.08% of the patients who received abnormal signals) were recommended to visit the hospital, with 182 (9.10%) showing specific details of cardiovascular diseases. The abnormalities detected included atrial fibrillation (AF), sinus arrhythmia (SA), premature atrial contraction (PAC), bundle branch block (BBB), atrioventricular (AV) block, premature ventricular contraction (PVC), ventricular fibrillation (VF), tachycardia, bradycardia, ST depression, chest pain, palpitations, and other abnormal rhythms. Notably, trial 13 had the highest number of abnormalities, with 40 cases (27.78% of its participants), including PVC, arrhythmia, and tachycardia, among others. The detailed cardiovascular disease findings indicated the presence of various types of arrhythmias and other heart conditions, with PVC being the most frequently observed abnormality.

### 3.3. Observations of Participants Who Were Recommended to Visit a Hospital

The 182 people who participated in the empirical research and received recommendations for cardiovascular disease-related treatment through abnormal signal monitoring were followed up through phone calls to determine whether they visited the hospital, the timing of the hospital visits, the diagnosis results, and the diagnosis details. A total of 139 (76.37%) answered the phone, and 30 (21.58% of those who answered the phone) visited the hospital and received treatment. Table 4 presents the results of the observations among the 30 participants who received treatment after visiting the hospital. Based on the date that each participant participated in the empirical research, 24 (80.00%) visited the hospital and underwent examination within one month. A total of 30 patients were analyzed, with the majority (56.67%) being diagnosed using only an electrocardiogram (EKG). A combined approach of EKG and ultrasound was utilized for 20% of the patients, while ultrasound alone accounted for 16.67%. Additionally, a small percentage were diagnosed using EKG and X-ray (3.33%) or general treatment methods (3.33%). Regarding the timing of hospital visits, 80% of the patients sought medical attention within one month of experiencing symptoms, whereas 20% delayed their visits beyond one month. The diagnosis results were meaningful as cardiovascular disease was discovered in seven people. In detail, progress observation (6.67%), stent procedures (3.33%), arrhythmia (3.33%), combined atrial fibrillation and arrhythmia (3.33%), heart medication prescriptions (3.33%), and panic disorder (3.33%) were included.

### 3.4. Satisfaction Survey after Completion of Empirical Research

Table 5 indicates the results of the participants’ satisfaction survey after the completion of the empirical research. A total of 1883 (94.15%) had not previously participated in clinical studies. A vast majority (96.55%) found the health information provided by the device, including ECG, heart rate, respiration, and body temperature, to be helpful. Most participants (97.95%) believed that the device would be beneficial for health management. In terms of comfort, 64.35% reported feeling ‘Very Satisfied’ or ‘Satisfied’, with an average satisfaction score of 2.02 ± 1.35. Some participants did experience discomfort related to detachment, itchiness, or restriction of movement. Trust in the health information provided by the devices was also strong, with 1510 participants (75.50%) indicating ‘Very Satisfied’ or ‘Satisfied’ and an average satisfaction score of 3.87 ± 1.20. Interest in the future use of the HiCardi device was moderate, with 48.20% of all showing ‘Very Satisfied’ or ‘Satisfied’ responses and an average satisfaction score of 3.53 ± 1.13. Regarding the ease of accessing health information through the mobile application, 75.95% of participants were ‘Very Satisfied’ or ‘Satisfied’, with an average satisfaction score of 3.88 ± 1.15. Overall satisfaction with the empirical research was high, with 78.65% of participants expressing ‘Very Satisfied’ or ‘Satisfied’ sentiments and an average satisfaction score of 3.98 ± 1.17.

## 4. Discussion

This study involved empirical research that remotely monitored single-lead ECG signals using a patch-type portable IoMT device. The device detected abnormal signals of atrial fibrillation (AF), arterial arrhythmia (SA), premature atrial contractions (PACs), bundle branch block (BBB), atrioventricular (AV) block, premature ventricular contractions (PVCs), ventricular fibrillation (VF), tachycardia, bradycardia, ST depression, chest pain, palpitations, and other abnormal rhythms. In this study, it was meaningful to test whether the remote IoMT monitoring system and device can be effective in detecting cardiovascular disease when monitoring daily activities. Paddy M. Barrett et al. compared 24 h Holter monitors with 14-day adhesive patch monitors in 146 patients; the patch monitor detected more arrhythmic events [25]. Furthermore, in the study by Warren M. Smith et al., using a P-wave-centric ECG patch monitor, rhythm changes requiring management alterations were detected in 46% of the 50 patients, compared to 12% with the Holter monitor [26]. These studies suggest that the IoMT device may offer higher diagnostic rates and convenience compared to traditional Holter monitors. In addition, Choi et al. focused on developing a low-cost, non-invasive heart monitoring system for early AF detection [17]. The device analyzed heartbeat variations and RR intervals in the ECG signal, generating alerts when abnormal patterns indicative of AF were detected. This real-time monitoring system was designed for home use, providing a practical solution for individuals with AF concerns. Similar to these previous studies, the detection of abnormal signals related to cardiovascular diseases using single-lead ECG portable devices has been extensively studied, showcasing the potential of these devices in everyday healthcare monitoring.

In this study, approximately 15 types of cardiovascular disease-related abnormal signals were observed using the IoMT device, including AF, SA, PACs, BBB, AV block, PVCs, VF, tachycardia, bradycardia, ST depression, chest pain, and palpitations. Among these, PVCs were the most frequently observed, with 79 instances. Examining other studies related to remote ECG monitoring, Markus Lueken et al. focused on the automated signal quality assessment of single-lead ECG recordings for the early detection of various arrhythmias, including PVCs [27]. This study reported that PVCs were observed in approximately 75% of participants during 24 to 48 h ambulatory ECG monitoring, demonstrating a high prevalence of these abnormal signals in the monitored population. Similarly, in the study by B. Venkataramanaiah et al. [28], the prevalence and characteristics of premature ventricular contractions (PVCs) were examined using remote ECG monitoring systems. This study emphasized the importance of detecting PVCs due to their potential to develop into more serious arrhythmias. The results indicated that PVCs were observed in 67.7% of the monitored population via 24 h Holter monitoring. Among these, 60.0% had a PVC burden of less than 5%, while 7.7% had a PVC burden of 5% or more. As shown in our study and in others, the occurrence rate of PVCs among abnormal ECG signals is relatively high with both IoMT devices and Holter monitoring systems. This is because PVCs can be precursors to various serious arrhythmias [29].

Additionally, our study found that 7 out of 30 patients (23%) who visited the hospital were diagnosed with actual heart disease using the IoMT device. This indicates that remote monitoring can have clinical effectiveness in preventative healthcare. A similar study by Marco V. Perez et al. involved 419,297 participants wearing Apple Watches, where those with detected irregular pulses acquired ECG patches for 7 days of monitoring [30]. Among the 2161 participants who received irregular pulse alarms, 450 returned ECG patches, and 34% of these showed atrial fibrillation. The positive predictive value of the irregular pulse alarms was 84%. The Fitbit Heart Study by Steven A. Lubitz et al. followed a similar approach, analyzing 455,699 participants [31]. Those with detected irregular heart rhythms were provided with an ECG patch for one week. The Fitbit PPG-based algorithm detected irregular heart rhythms and used ECG patches to confirm atrial fibrillation. The results indicated a high positive predictive value of 98%, demonstrating the effectiveness of consumer wearable technology in identifying AF episodes. Similar to these two studies, remote monitoring could be expected to have clinical effects on preventive health management in this study. However, it had the limitation that the sample size used in the empirical research was small, making it difficult to achieve representativeness. In addition, since the empirical research was conducted in a designated location, factors such as sweat, shortness of breath, or movement during the hike, which may have caused poor contact with the patch, and the geographical characteristics of the mountain potentially affected the transmission of the ECG signal, which may have affected the results.

Lastly, a limitation of this study is that the compliance rate among those who responded to the recommendations for hospital treatment after discovering abnormal signs suggestive of cardiovascular disease and actually visiting the hospital was low. Similarly, many other studies also face low compliance with follow-up calls. For example, in a study by Dorothy M. Mwachiro et al., nurses conducted follow-up calls to evaluate their effectiveness in reducing 30-day readmissions post-discharge in neurosurgery patients [32]. However, only 45% of the 83 patients received follow-up calls, impacting the reliability of the study’s assessment of readmission rates and post-discharge care. Another study by Phil Edwards investigated the challenges of long-term follow-up data collection in non-commercial, academically led breast cancer clinical trials [33]. Based on the UK perspective, the study highlighted difficulties in patient compliance and data collection in long-term follow-up studies. The low response rates made data collection challenging, affecting the reliability of the study outcomes. The researchers suggested that maintaining continuous contact with patients plays a crucial role in securing long-term follow-up data. Comparing these studies, our research also faced challenges with follow-up calls, which may have limited the study’s results. If a higher response rate had been achieved, the diagnostic accuracy might have exceeded 23%. This highlights the difficulty of data collection through follow-up calls and indicates the need for new approaches to address these challenges in future research.

## 5. Conclusions

We conducted a pilot study on the remote monitoring of ECG signals measured by a patch-type portable IoMT device targeting 2000 ordinary people. In a situation where face-to-face medical treatment by a doctor was the legal principle, it was thought that this intervention had great implications in that it contributed to preventive health management by detecting cardiovascular diseases in the daily life of ordinary people under conditions where non-face-to-face remote monitoring was temporarily permitted. However, this study was limited in that it emphasized remote monitoring and reported somewhat descriptive analysis results. In future studies, it would be necessary to propose a model that predicts or learns cardiovascular diseases by analyzing ECG signals with artificial intelligence. Furthermore, to validate the practical effectiveness of the data collected in this study, we plan to conduct an in-depth kinetic study through a metabolic equivalent (MET) analysis from a kinematic perspective as further research. This is significant as a more convenient method for monitoring and diagnosing cardiovascular diseases compared to treadmill tests conducted in hospitals. It is also expected to contribute to preventative healthcare by enabling immediate actions such as hospital visits or medication intake.

## Figures and Tables

**Figure 1 bioengineering-11-00836-f001:**
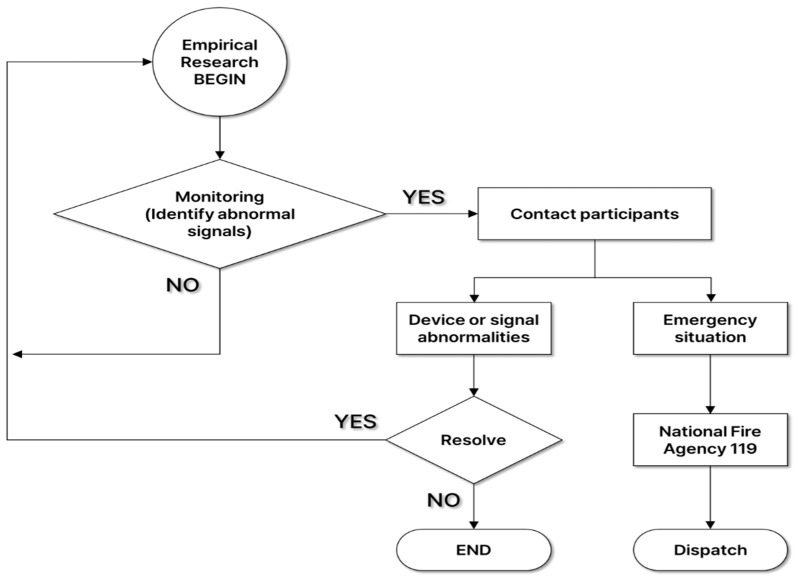
Emergency response protocols in place.

**Figure 2 bioengineering-11-00836-f002:**
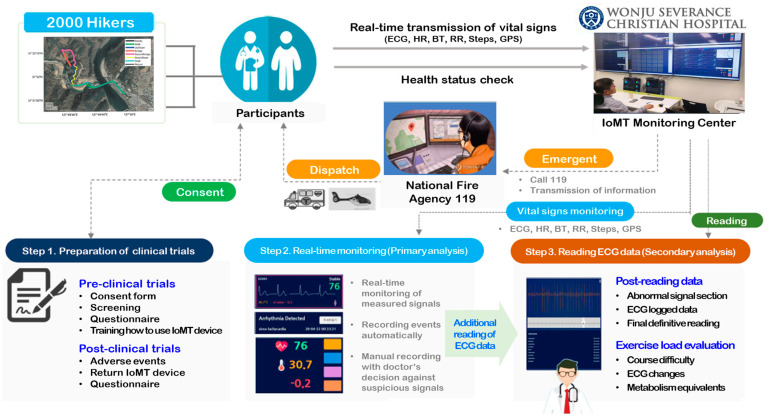
Flowchart of IoMT monitoring service.

**Table 1 bioengineering-11-00836-t001:** Regulation evidences and substantiation.

Regulation Evidence	Substantiation
Medical service act	- Diabetes management service based on medical information - Blood pressure management service based on medical information - Chronic disease from-home monitoring service - Biosignal monitoring healthcare service
Article 34 (Remote Medical Treatment)	
Medical service act	- Portable X-ray diagnosis system
Article 37 (Radiation Generating Devices for Diagnostic)	
Pharmaceutical affairs act	- Vaccine demand prediction service (DUR demonstration)
Article 23-3 (Establishment and Operation of Information System for Safe Use of Drugs)	

**Table 2 bioengineering-11-00836-t002:** Demographic and health characteristics of participants.

Variables	Participants (Hikers)		
	Total (*n* = 2000)	Male (*n* = 811)	Female (*n* = 1189)
Age (mean ± S.D)	49.36 ± 14.65	49.35 ± 15.81	49.37 ± 13.81
Age group (*n*, %)			
Under 20	51 (2.55%)	22 (2.71%)	29 (2.43%)
20s	248 (12.40%)	118 (14.54%)	130 (10.93%)
30s	266 (13.30%)	105 (12.94%)	161 (13.54%)
40s	353 (17.65%)	133 (16.39%)	220 (18.50%)
50s	561 (28.05%)	188 (23.18%)	373 (31.37%)
60s	428 (21.40%)	187 (23.05%)	241 (20.26%)
More than 70	93 (4.65%)	58 (7.15%)	35 (2.94%)
Height (cm)	164.16 ± 8.65	171.35 ± 7.14	159.26 ± 5.63
Weight (kg)	64.26 ± 11.80	73.04 ± 10.97	58.26 ± 7.98
SBP (mmHg)	130.89 ± 17.12	136.85 ± 14.36	126.82 ± 17.65
DBP (mmHg)	85.61 ± 10.59	88.79 ± 10.43	83.44 ± 10.15
PR (bpm)	74.87 ± 22.94	76.05 ± 33.77	74.07 ± 10.30
Temperature (°C)	36.72 ± 10.32	36.79 ± 11.49	36.67 ± 9.44
Chronic disease (*n*, %)			
None	1325 (66.25%)	518 (63.87%)	807 (67.87%)
One	422 (21.10%)	183 (22.57%)	239 (20.10%)
Two	187 (9.35%)	77 (9.49%)	110 (9.25%)
More than three	66 (3.30%)	33 (4.07%)	33 (2.78%)
Family history (*n*, %)			
Yes	906 (45.30%)	297 (36.62%)	609 (51.22%)
No	1094 (54.70%)	514 (63.38%)	580 (48.78%)

S.D: standard deviation; SBP: systolic blood pressure; DBP: diastolic blood pressure; PR: pulse rate.

**Table 3 bioengineering-11-00836-t003:** Summary of clinical trial participation and ECG abnormalities.

Trials	Participants	Abnormal Signals	Recommendation to Visit a Hospital		Details about CD (n)
			Total	CD	
1	21	-	-	-	-
2	10	-	-	-	-
3	25	3	3	2	AF (1), SA (1)
4	44	1	1	1	PVC (1)
5	58	4	2	1	Arrhythmia (1)
6	59	5	5	3	AF (2), Tachycardia (1)
7	37	13	11	2	Arrhythmia (1), Bradycardia (1)
8	65	8	7	5	Arrhythmia (2), SA block (1), PAC (1), BBB (1)
9	46	15	15	4	AV block (2), Tachycardia (1), Bradycardia (1)
10	26	6	4	1	Tachycardia (1)
11	8	-	-	-	-
12	141	25	22	2	Arrhythmia (1), Tachycardia (1)
13	144	40	36	21	PVC (7), Arrhythmia (4), Tachycardia (3), AF (2), PAC (1), Chest pain (2), Palpitations (1), Abnormal rhythms (1)
14	48	9	8	4	PVC (2), Arrhythmia (2)
15	48	9	7	3	AF (1), Arrhythmia (1), PVC (1)
16	48	5	5	4	PVC (2), VF (1), BBB (1)
17	48	10	10	9	Arrhythmia (3), Bradycardia (2), AF (1), PVC (1) BBB (1), ST depression (1)
18	144	23	23	17	PVC (9), BBB (5), AF (2), Tachycardia (1)
19	144	16	16	14	PVC (10), Arrhythmia (2), BBB (1), Sinus arrhythmia (1)
20	48	10	10	7	BBB (3), Arrhythmia (2), PVC (1), PAC (1)
21	48	9	9	5	Arrhythmia (3), AF (1), PVC (1)
22	82	17	15	7	PVC (6), BBB (1)
23	72	10	9	6	Arrhythmia (3), PVC (2), BBB (1)
24	72	8	8	6	PAC (2), PVC (1), Arrhythmia (1), AF (1), Abnormal rhythms (1)
25	48	12	12	7	PVC (5), AF (1), PAC (1)
26	72	13	13	12	PVC (8), BBB (2), Arrhythmia (1), Tachycardia (1)
27	168	22	22	22	PVC (15), PAC (5), Arrhythmia (1), Tachycardia (1)
28	48	4	4	3	BBB (3)
29	48	9	9	5	PVC (2), Arrhythmia (2), PAC (1)
30	130	12	10	9	PVC (5), Arrhythmia (3), Tachycardia (1)
Total	2000	318	296	182	

CD: cardiovascular disease; AF: atrial fibrillation; SA: sinus arrhythmia; SA block: sinoatrial block; PAC: premature atrial contraction; BBB: bundle branch block; AV block: atrioventricular block; PVC: premature ventricular contraction; VF: ventricular fibrillation.

**Table 4 bioengineering-11-00836-t004:** Observations of hospital visits by participants with abnormal ECG signals.

Variables	Number of Patients
Diagnosis method (*n*, %)	
EKG	17 (56.67)
EKG and ultrasound	6 (20.00)
Ultrasound	5 (16.67)
EKG and X-ray	1 (3.33)
General treatment	1 (3.33)
Timing of hospital visits (*n*, %)	
Within one month	24 (80.00)
After one month	6 (20.00)
Diagnosis results (*n*, %)	
Progress observation	2 (6.67)
Stent procedure	1 (3.33)
Arrhythmia	1 (3.33)
AF and arrhythmia	1 (3.33)
Heart medication prescription	1 (3.33)
Panic disorder	1 (3.33)
No abnormality	23 (76.68)

EKG: electrocardiogram; AF: atrial fibrillation.

**Table 5 bioengineering-11-00836-t005:** Results of participants’ satisfaction survey after completion of empirical research.

Questions	Values	
	Distribution (*n*, %)	Score (Mean ± S.D)
Have you previously participated in any clinical studies?		
Yes	117 (5.85)	-
No	1883 (94.15)	
Was the health information provided helpful? (e.g., ECG, heart rate, respiration, body temperature)		
Yes	1931 (96.55)	-
No	69 (3.45)	
Do you believe that the devices used will be helpful for health management?		
Yes	1959 (97.95)	-
No	41 (2.05)	
Did you feel any discomfort after attaching the device? (e.g., detachment, itchiness, restriction of movement)		
Very Satisfied (1)	802 (40.10)	2.02 ± 1.35
Satisfied (2)	485 (24.25)	
Neutral (3)	39 (1.95)	
Dissatisfied (4)	123 (6.15)	
Very Dissatisfied (5)	187 (9.35)	
Do you trust the health information provided? (e.g., ECG, heart rate, respiration, body temperature)		
Very Satisfied (5)	700 (35.00)	3.87 ± 1.20
Satisfied (4)	810 (40.50)	
Neutral (3)	174 (8.70)	
Dissatisfied (2)	161 (8.05)	
Very Dissatisfied (1)	155 (7.75)	
Are you interested in using the HiCardi (attached device) in the future?		
Very Satisfied (5)	328 (16.40)	3.53 ± 1.13
Satisfied (4)	636 (31.80)	
Neutral (3)	362 (18.10)	
Dissatisfied (2)	205 (10.25)	
Very Dissatisfied (1)	105 (5.25)	
Is it easy to access health information through the mobile application? (e.g., ECG, heart rate, respiration, body temperature)		
Very Satisfied (5)	673 (33.65)	3.88 ± 1.15
Satisfied (4)	846 (42.30)	
Neutral (3)	187 (9.35)	
Dissatisfied (2)	163 (8.15)	
Very Dissatisfied (1)	131 (6.55)	
How satisfied are you with this empirical research?		
Very Satisfied (5)	813 (40.65)	3.98 ± 1.17
Satisfied (4)	760 (38.00)	
Neutral (3)	164 (8.20)	
Dissatisfied (2)	117 (5.85)	
Very Dissatisfied (1)	146 (7.30)	

EKG: electrocardiogram; S.D: standard deviation.

## Data Availability

The data that support the findings of this study are available on request from the corresponding author. The data are not publicly available due to privacy or ethical restrictions.

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
