# Peer review of "Substantiation and Effectiveness of Remote Monitoring System Based on IoMT Using Portable ECG Device"

_bioengineering, 2024, doi:10.3390/bioengineering11080836_

Round 1

Reviewer 1 Report

Comments and Suggestions for Authors

The authors introduced an efficient framework based on IoT devices, particularly HiCardi wearable sensors, to monitor and assess the hiking activities of 2000 participants. Generally, the paper is well organized and easy to follow. The performed statistics are important and insightful. However, the proposed study suffers from several drawbacks that should be addressed by the authors:

1) The major contributions should be mentioned clearly at the end of the Introduction section.

2) Please mention the inclusion and exclusion criteria in the medical experiment.

3) I miss a related work section to describe similar studies along with their shortcomings.

4) The research aspect is totally missed in this study. The authors only performed descriptive analysis of the collected data without proposing new predictive or learning models.

5) A major weakness of the paper is that the authors did not compare their framework to similar existing ones.

6) Reformulate the conclusion in a way to show the main findings of the study as well as the limitations.  

7) The number of used references is embarrassing. I advise to add more recent references. 

Author Response

※ Comments and Suggestions for Authors

The authors introduced an efficient framework based on IoT devices, particularly HiCardi wearable sensors, to monitor and assess the hiking activities of 2000 participants. Generally, the paper is well organized and easy to follow. The performed statistics are important and insightful. However, the proposed study suffers from several drawbacks that should be addressed by the authors:

1) The major contributions should be mentioned clearly at the end of the Introduction section.

☞ We appreciate your sincere comments. We have tried to supplement the Introduction section with the background and importance of conducting this study.

☞ [Revised] (Line 39-169) In South Korea, remote monitoring in healthcare encounters significant legal obstacles due to the Medical Service Act, particularly Article 34, which prohibits remote diagnosis and treatment between doctors and patients [1]. This article mandates that medical services must be delivered face-to-face, emphasizing the necessity of in-person interactions for diagnosis and treatment. Although the first telemedicine pilot project was attempted in 1988, the debate over its introduction continues today. On the one hand, promoting and activating the telemedicine business along with the development of information and communication technology is an important task for the country in terms of industrial entry, and urges the promotion of the project, saying that science and technology will benefit the public by providing quality medical services. On the other hand, they say that medical practice that does not require face-to-face contact is still risky and that there is no need to pursue this project while taking such risks [2]. As a result, the law restricts the growth of telemedicine and remote monitoring technologies within the country.

Conversely, countries like the United States and those in the European Union have adopted more liberal legal frameworks and economic incentives, which facilitate the rapid development and widespread integration of remote monitoring technologies into their healthcare systems. These nations have embraced telemedicine and remote monitoring, recognizing their potential to enhance healthcare access, improve patient outcomes, and reduce costs. In the United States, remote monitoring has been extensively utilized to manage various health conditions, including COVID-19, chronic obstructive pulmonary disease (COPD), and heart failure. In Japan, remote monitoring is also being applied in diverse healthcare settings, including neonatal care and immunization safety monitoring. In the United States, the Providence health system implemented a Home Monitoring Program (HMP) for COVID-19 patients in March 2020. This program utilized at-home pulse oximeters, thermometers, and text-based surveys to monitor symptoms. The HMP was found to be highly effective, with over 80% engagement in daily text-based surveys and high levels of comfort using home monitoring devices among both English and Spanish-speaking participants. Additionally, the program was associated with increased outpatient and emergency department encounters, indicating that patients felt safer and more satisfied monitoring their condition from home [3]. Similarly, remote patient monitoring for chronic diseases such as COPD has shown potential benefits, including early detection of exacerbations, prompt access to therapy, and ultimately improved patient outcomes [4]. In heart failure management, telemonitoring strategies became routine during the pandemic, with evidence supporting the efficacy and safety of virtual visits and remote vital sign monitoring. The intensity of monitoring was tailored to match the patient's risk profile, highlighting the importance of a structured approach to telehealth implementation [5]. In Japan, remote monitoring is utilized in various healthcare contexts, including the monitoring of adverse events following immunization (AEFI). The Japanese AEFI monitoring system ensures the safe implementation of immunizations by tracking and addressing adverse events. This system is complemented by a unique compensation mechanism, known as the relief system, which provides support for individuals affected by immunization-related issues [6]. Additionally, remote monitoring is employed in neonatal intensive care units (NICUs) in Japan. The use of transcutaneous (tc) measurements of partial pressure of oxygen (tcPO2) and carbon dioxide (tcPCO2) in NICUs allows for continuous and non-invasive monitoring of critically ill neonates, ensuring timely interventions and better health outcomes [7]. As such, Korea has a problem that reduces market dynamism due to positive regulations based on laws that list only those things that are permitted, thereby hindering the growth of the industry along with the flow of the Fourth Industrial Revolution [8].

As a measure to solve this problem, the Ministry of SMEs and Startups of Korea has established a system to designate the Regulation Free Special Zone (RFZ) by region and to ease key regulations related to new businesses as a package, in order to create an environment that can focus on rapidly changing technological industrialization in the era of the 4th Industrial Revolution [9]. First of all, a study analyzing the legal and institutional issues of regulations recognized as obstacles in industrial sites was reported [10-12], and in 2019, the Gangwon Digital Healthcare Regulation Free Special Zone was designated to make it grow the digital healthcare industry, making it possible for Gangwon State to conduct empirical research as a test bed for regulatory improvement of various laws, as shown in Table 1 below.

Table 1. Regulation evidences and substantiation.

Regulation Evidence

Substantiation

MEDICAL SERVICE ACT

-          Diabetes management service based on medical information

-          Blood pressure management service based on medical information

-          Chronic disease from home monitoring service

-          Bio-signal monitoring healthcare service

Article 34 (Remote Medical Treatment)

MEDICAL SERVICE ACT

-          Portable X-ray diagnosis system

Article 37 (Radiation Generating Devices for Diagnostic)

PHARMACEUTICAL AFFAIRS ACT

-          Vaccine demand prediction service (DUR demonstration)

Article 23-3 (Establishment and Operation of Information System for Safe Use of Drugs)

In this study, in order to ease the regulations under Article 34 of the Medical Act of Korea, a bio-signal monitoring healthcare service was planned that partially applied that wearable devices for bio-signal monitoring would be provided and that a doctor in the hospital would be able to monitor electrocardiogram (ECG) for rapid treatment and rescue in emergency situations. Then, an empirical study was conducted on remote monitoring between a doctor and patients depending on the limited scope, location, and method. ECG is a primary diagnostic tool for various cardiovascular diseases, including arrhythmias, myocardial infarctions, and other cardiac abnormalities. The high diagnostic accuracy of ECGs in detecting myocardial ischemia was demonstrated, reinforcing the critical role of ECGs in early detection and treatment planning for cardiovascular conditions [13]. Also, studies such as those by Mincholé and Rodríguez have highlighted the predictive power of ECG signals in identifying patients at risk of future cardiac events, further underscoring the importance of ECG-based diagnostics [14]. In addition, remote monitoring of ECG signals has become increasingly important, especially with the rise of chronic heart disease and the need for continuous, real-time monitoring. A report by the American Heart Association (AHA) in 2019 emphasized that remote monitoring can significantly improve patient outcomes by enabling early detection of potential issues, thereby allowing for timely medical interventions [15]. It was found that remote monitoring of heart failure patients reduced hospitalizations and improved survival rates. These findings highlight the critical role of remote ECG monitoring in managing cardiovascular health, especially in at-risk populations [16].

However, despite the various studies reported on ECG signal monitoring, some shortcomings may affect its efficacy and reliability. These shortcomings can be broadly categorized into device power, signal quality, and practical use. One of the primary disadvantages is related to the power constraints of ECG monitoring devices, particularly implantable ones. The limited battery capacity restricts the functionalities of these devices, preventing them from achieving their full potential. Although wireless power transfer (WPT) technology has been proposed to address this issue, the implementation and efficiency of such systems in real-world scenarios remain to be thoroughly validated [17]. Signal quality is another significant challenge in ECG monitoring. During data acquisition and transmission, various artifacts can degrade the ECG signal, leading to incorrect clinical diagnoses. These artifacts may stem from environmental noise, patient movement, or device-related issues. While advanced signal processing techniques such as wavelet transforms, neural networks, and adaptive filters have been developed to suppress these artifacts, ensuring consistently high-quality signals remains a complex task. The proposed multistage adaptive filter aims to tackle multiple types of artifacts, but it requires prior knowledge of the interferences, which may not always be feasible [18]. Wearable ECG devices, such as patch-type monitors, introduce their own set of challenges. These devices, while unobtrusive and user-friendly, are typically suited for short to medium-term monitoring (days to weeks) rather than long-term continuous use. Moreover, achieving a balance between low power consumption and high diagnostic accuracy in these devices remains a significant challenge. Existing schemes often suffer from high power consumption to maintain accuracy, which can limit their practicality [19]. Another critical issue is the difficulty in ensuring the clinical acceptability of ECG signals in unsupervised environments. Real-time signal quality assessment (SQA) methods have been proposed to classify the acquired ECG signal into acceptable or unacceptable categories, but these methods need to be robust and efficient to be effective in practical applications [20].

The Internet of Medical Things (IoMT) plays a crucial role in modern healthcare by facilitating remote monitoring. It enables the collection and analysis of patients' vital signs and health data in real-time, allowing healthcare professionals to continuously observe patients' conditions. This improves the efficiency of patient management and allows for quick responses in emergencies, ultimately enhancing patient safety and treatment outcomes [21]. It is a term for medical objects that apply IoT technology to the medical field and can transmit medical information and bio-signal data in real time through network communication. IoMT device includes an integrated circuit for data collection, a signal processing device for data processing, and a network module through data transmission. So, then this system can make a minimal human intervention and effectively monitor things that were not monitored before. Digital health, particularly involving the IoMT devices, is revolutionizing health management in daily life. Lupton reported that it was discussed how IoMT devices, such as wearable ECG monitors, provide continuous health data that can be analyzed to offer personalized health insights and interventions [22]. The advantages of using IoMT devices include improved patient adherence to treatment plans, real-time health monitoring, and the ability to collect large datasets for advanced analytics, which can lead to more accurate predictive models and personalized healthcare strategies. Also, it was demonstrated that digital health technologies not only improve health outcomes but also enhance the quality of life by allowing individuals to manage their health proactively and conveniently [23].

This study aimed to verify the effectiveness of remote biological signal monitoring technology using the IoMT device.

2) Please mention the inclusion and exclusion criteria in the medical experiment.

☞ Thank you for your opinion. We mentioned the inclusion and exclusion criteria in the paragraph, ‘2.1. Participants’..

☞ [Previous] (Line 76-80) In this study, a clinical trial was conducted to verify the effectiveness of the remote monitoring diagnosis technology using the IoMT device. The clinical trial was conducted a total of 30 times in about 8 months and a total of 2,000 hikers participated. Each hiker filled out a survey about the clinical trial consent and existing health information in advance then attached a wearable ECG monitoring patch to their chest and climbed the mountain.

   [Revised] (Line 174-180) In this study, a clinical trial was conducted to verify the effectiveness of the remote monitoring diagnosis technology using the IoMT device. The clinical trial was conducted a total of 30 times in about 8 months and a total of 2,000 hikers participated. The inclusion criteria were adults aged 19 years or older, who gave written consent to participate in the empirical research, and who could understand and follow the on-site training and instructions. Also, the exclusion criteria were those who could not attach the patch-type IoMT device due to skin disease, those who were pregnant, those who were participating in another clinical trial at the same time, or those who had recently completed participation in a previous clinical trial. Each hiker filled out a survey about the clinical trial consent and existing health information in advance then attached a wearable ECG monitoring patch to their chest and climbed the mountain.

3) I miss a related work section to describe similar studies along with their shortcomings.

☞ Thank you for your good comments We tried to add similar studies along with the shortcomings of the patch-type ECG device and remote monitoring in the Introduction section.

☞ [Revised] (Line 122-147) However, despite the various studies reported on ECG signal monitoring, some shortcomings may affect its efficacy and reliability. These shortcomings can be broadly categorized into device power, signal quality, and practical use. One of the primary disadvantages is related to the power constraints of ECG monitoring devices, particularly implantable ones. The limited battery capacity restricts the functionalities of these devices, preventing them from achieving their full potential. Although wireless power transfer (WPT) technology has been proposed to address this issue, the implementation and efficiency of such systems in real-world scenarios remain to be thoroughly validated [17]. Signal quality is another significant challenge in ECG monitoring. During data acquisition and transmission, various artifacts can degrade the ECG signal, leading to incorrect clinical diagnoses. These artifacts may stem from environmental noise, patient movement, or device-related issues. While advanced signal processing techniques such as wavelet transforms, neural networks, and adaptive filters have been developed to suppress these artifacts, ensuring consistently high-quality signals remains a complex task. The proposed multistage adaptive filter aims to tackle multiple types of artifacts, but it requires prior knowledge of the interferences, which may not always be feasible [18]. Wearable ECG devices, such as patch-type monitors, introduce their own set of challenges. These devices, while unobtrusive and user-friendly, are typically suited for short to medium-term monitoring (days to weeks) rather than long-term continuous use. Moreover, achieving a balance between low power consumption and high diagnostic accuracy in these devices remains a significant challenge. Existing schemes often suffer from high power consumption to maintain accuracy, which can limit their practicality [19]. Another critical issue is the difficulty in ensuring the clinical acceptability of ECG signals in unsupervised environments. Real-time signal quality assessment (SQA) methods have been proposed to classify the acquired ECG signal into acceptable or unacceptable categories, but these methods need to be robust and efficient to be effective in practical applications [20].

4) The research aspect is totally missed in this study. The authors only performed descriptive analysis of the collected data without proposing new predictive or learning models.

☞ Thank you for your serious point. However, we would like to explain that this study was composed of empirical research that attempted to ease the critical restrictions and regulations of our country's Medical Service Act. This study has implications in that in our country where face-to-face treatment by doctors is the rule, the regulations were temporarily relaxed to read abnormal ECG signals through non-face-to-face remote monitoring targeting the general public, inducing hospital visits, and actually diagnosing cardiovascular disease, which led to cases where it was possible to prevent it. Absolutely, we agreed with your opinion that if you understood the content of the study to predict cardiovascular disease by monitoring 2000 people's ECG signals, you should have come up with a new prediction or learning model. We are well aware of the limitations of this study and will strive to produce good results in future studies. We also have revised the Conclusion paragraph to reflect this content.

☞ [Previous] (Line 311-320) We demonstrated the effectiveness of remote cardiovascular monitoring during daily activities using IoMT devices by diagnosing actual heart disease with 23% accuracy through monitoring abnormal ECG signals collected from 2,000 general hikers using the patch-type electrocardiograph, HiCardi. Furthermore, to validate the practical effectiveness of the data collected in this study, we plan to conduct an in-depth kinetic study through the metabolic equivalents (METs) analysis from a kinematic perspective as a further study. This is significant as a more convenient method for monitoring and diagnosing cardiovascular diseases compared to treadmill tests conducted in hospitals. It is also expected to contribute to preventative healthcare by enabling immediate actions such as hospital visits or medication intake.

[Revised] (Line 433-447) We conducted a pilot study on remote monitoring of ECG signals measured by a patch-type portable IoMT device targeting 2,000 ordinary people. In a situation where face-to-face medical treatment by a doctor was the legal principle, it was thought that it has great implications in that it has contributed to preventive health management by detecting cardiovascular diseases in the daily life of ordinary people under conditions where non-face-to-face remote monitoring was temporarily permitted. However, this study was limited in that it emphasized remote monitoring and reported somewhat descriptive analysis results. In future studies, it would be necessary to propose a model that predicts or learns cardiovascular diseases by analyzing ECG signals with artificial intelligence. Furthermore, to validate the practical effectiveness of the data collected in this study, we plan to conduct an in-depth kinetic study through the metabolic equivalents (METs) analysis from a kinematic perspective as a further study. This is significant as a more convenient method for monitoring and diagnosing cardiovascular diseases compared to treadmill tests conducted in hospitals. It is also expected to contribute to preventative healthcare by enabling immediate actions such as hospital visits or medication intake.

5) A major weakness of the paper is that the authors did not compare their framework to similar existing ones.

☞ Thank you for pointing out the weakness of this paper. We tried to compensate for this weakness by adding existing studies similar to the framework of this study.

☞ [Previous] (Line 243-309) This study conducted a clinical trial on 2,000 ordinary hikers to verify the effectiveness of the remote monitoring technology using the IoMT devices and system. As a result, 7 of 2,000 hikers in real-time using the IoMT devices and system were real-diagnosed with heart diseases. In the case of some heart disease diagnoses, many cases are difficult to detect in the existing monitoring system, and monitoring for more than a week is required in some cases. In this study, it was meaningful to test that the remote IoMT monitoring system and device in daily activities can be effective in detecting heart disease. Similarly, in the study by Paddy M. Barrett et al [9], comparing 24-hour Holter monitors with 14-day adhesive patch monitors in 146 patients, the patch monitor detected more arrhythmic events. Furthermore, in the study by Warren M. Smith et al [10], using a P-wave-centric ECG patch monitor, 46% of the 50 patients detected rhythm changes requiring management alterations, compared to 12% with the Holter monitor. These studies suggest that the IoMT device may offer higher diagnostic rates and convenience compared to traditional Holter monitors.

In this study, approximately 15 types of heart disease-related abnormal signals were observed using the IoMT device, including PVCs and atrial fibrillation. Among these, PVCs were the most frequently observed, with 79 instances. Examining other studies related to remote ECG monitoring, Markus Lueken et al [11] focused on the automated signal quality assessment of single-lead ECG recordings for the early detection of various arrhythmias, including PVCs. This study reported that PVCs were observed in approximately 75% of participants during 24 to 48-hour ambulatory ECG monitoring, demonstrating a high prevalence of these abnormal signals in the monitored population. Similarly, in the study by B. Venkataramanaiah et al [12], the prevalence and characteristics of premature ventricular contractions (PVCs) were examined using remote ECG monitoring systems. This study emphasized the importance of detecting PVCs due to their potential to develop into more serious arrhythmias. The results indicated that PVCs were observed in 67.7% of the monitored population via 24-hour Holter monitoring. Among these, 60.0% had a PVC burden of less than 5%, while 7.7% had a PVC burden of 5% or more. As shown in our study and others, the occurrence rate of PVCs among abnormal ECG signals is relatively high in both IoMT devices and Holter monitoring systems. This is because PVCs can be precursors to various serious arrhythmias [13].

Additionally, our study found that 7 out of 30 patients (23%) who visited the hospital were diagnosed with actual heart disease using the IoMT device. This indicates that remote monitoring can have clinical effectiveness in preventative healthcare. A similar study by Marco V. Perez et al [14] involved 419,297 participants wearing Apple Watches, where those with detected irregular pulses acquised ECG patches for 7 days of monitoring. Among the 2,161 participants who received irregular pulse alarms, 450 returned ECG patches, and 34% of these showed atrial fibrillation. The positive predictive value of the irregular pulse alarms was 84%. The Fitbit Heart Study by Steven A. Lubitz et al [15] followed a similar approach, analyzing 455,699 participants. Those with detected irregular heart rhythms were provided with an ECG patch for one week. The Fitbit PPG-based algorithm detected irregular heart rhythms and used ECG patches to confirm atrial fibrillation. Results indicated a high positive predictive value of 98%, demonstrating the effectiveness of consumer wearable technology in identifying AF episodes. Compared to these two studies, our study's diagnostic accuracy appears relatively lower. However, this could be attributed to factors such as sweat, heavy breathing, or movement during hiking causing poor contact with the patch, and geographic characteristics of being in the mountains potentially affecting the transmission of ECG signals. While our study's accuracy of diagnosis is relatively lower, this can be attributed to the limitation of a small number of confirmed follow-up calls.

Moreover, a small number of confirmed follow-up calls is a limitation of our study. Many other studies also face low compliance with follow-up calls. For example, in a study by Dorothy M. Mwachiro et al [16], nurses conducted follow-up calls to evaluate their effectiveness in reducing 30-day readmissions post-discharge in neurosurgery patients. However, only 45% of the 83 patients received follow-up calls, impacting the reliability of the study's assessment of readmission rates and post-discharge care. Another study by Phil Edwards [17] investigated the challenges of long-term follow-up data collection in non-commercial, academically-led breast cancer clinical trials. Based on the UK perspective, the study highlighted difficulties in patient compliance and data collection in long-term follow-up studies. Low response rates made data collection challenging, affecting the reliability of the study outcomes. The researchers suggested that maintaining continuous contact with patients plays a crucial role in securing long-term follow-up data. Comparing these studies, our research also faced challenges with follow-up calls, which may have limited the study's results. If a higher response rate had been achieved, the diagnostic accuracy might have exceeded 23%. This highlights the difficulty of data collection through follow-up calls and indicates the need for new approaches to address these challenges in future research.

[Revised] (Line 354-431) This study was an empirical research that remotely monitored single-lead ECG signals using a patch-type portable IoMT device. The device detected abnormal signals of atrial fibrillation (AF), arterial arrhythmia (SA), premature atrial contractions (PAC), bundle branch block (BBB), atrioventricular (AV) block, premature ventricular contractions (PVC), ventricular fibrillation (VF), tachycardia, bradycardia, ST depression, chest pain, palpitations, and other abnormal rhythms. In this study, it was meaningful to test that the remote IoMT monitoring system and device in daily activities can be effective in detecting cardiovascular disease. Paddy M. Barrett et al. compared 24-hour Holter monitors with 14-day adhesive patch monitors in 146 patients, the patch monitor detected more arrhythmic events [25]. Furthermore, in the study by Warren M. Smith et al., using a P-wave-centric ECG patch monitor, 46% of the 50 patients detected rhythm changes requiring management alterations, compared to 12% with the Holter monitor [26]. These studies suggest that the IoMT device may offer higher diagnostic rates and convenience compared to traditional Holter monitors. In addition, Choi et al. focused on developing a low-cost, non-invasive heart monitoring system for early AF detection [17]. The device analyzed heartbeat variations and RR intervals in the ECG signal, generating alerts when abnormal patterns indicative of AF were detected. This real-time monitoring system was designed for home use, providing a practical solution for individuals with AF concerns. Similar to these previous studies, the detection of abnormal signals related to cardiovascular diseases using single-lead ECG portable devices has been extensively studied, showcasing the potential of these devices in everyday healthcare monitoring.

In this study, approximately 15 types of cardiovascular disease-related abnormal signals were observed using the IoMT device, including AF, SA, PACs, BBB, AV block, PVCs, VF, tachycardia, bradycardia, ST depression, chest pain, and palpitations. Among these, PVCs were the most frequently observed, with 79 instances. Examining other studies related to remote ECG monitoring, Markus Lueken et al. focused on the automated signal quality assessment of single-lead ECG recordings for the early detection of various arrhythmias, including PVCs [27]. This study reported that PVCs were observed in approximately 75% of participants during 24 to 48-hour ambulatory ECG monitoring, demonstrating a high prevalence of these abnormal signals in the monitored population. Similarly, in the study by B. Venkataramanaiah et al [28], the prevalence and characteristics of premature ventricular contractions (PVCs) were examined using remote ECG monitoring systems. This study emphasized the importance of detecting PVCs due to their potential to develop into more serious arrhythmias. The results indicated that PVCs were observed in 67.7% of the monitored population via 24-hour Holter monitoring. Among these, 60.0% had a PVC burden of less than 5%, while 7.7% had a PVC burden of 5% or more. As shown in our study and others, the occurrence rate of PVCs among abnormal ECG signals is relatively high in both IoMT devices and Holter monitoring systems. This is because PVCs can be precursors to various serious arrhythmias [29].

Additionally, our study found that 7 out of 30 patients (23%) who visited the hospital were diagnosed with actual heart disease using the IoMT device. This indicates that remote monitoring can have clinical effectiveness in preventative healthcare. A similar study by Marco V. Perez et al. involved 419,297 participants wearing Apple Watches, where those with detected irregular pulses acquired ECG patches for 7 days of monitoring [30]. Among the 2,161 participants who received irregular pulse alarms, 450 returned ECG patches, and 34% of these showed atrial fibrillation. The positive predictive value of the irregular pulse alarms was 84%. The Fitbit Heart Study by Steven A. Lubitz et al. followed a similar approach, analyzing 455,699 participants [31]. Those with detected irregular heart rhythms were provided with an ECG patch for one week. The Fitbit PPG-based algorithm detected irregular heart rhythms and used ECG patches to confirm atrial fibrillation. Results indicated a high positive predictive value of 98%, demonstrating the effectiveness of consumer wearable technology in identifying AF episodes. Similar to these two studies, remote monitoring could be expected to have clinical effects on preventive health management in this study. However, it had the limitation that the sample size of the empirical research was small, making it difficult to have representativeness. In addition, since the empirical research was conducted in a designated location, factors such as sweat, shortness of breath, or movement during the hike, which may have caused poor contact with the patch, and the geographical characteristics of the mountain potentially affected the transmission of the ECG signal, which may have affected the results.

Lastly, a limitation of this study is that the compliance rate among those who responded to the recommendations for hospital treatment after discovering abnormal signs suggestive of cardiovascular disease and actually visiting the hospital was low. Similarly, many other studies also face low compliance with follow-up calls. For example, in a study by Dorothy M. Mwachiro et al., nurses conducted follow-up calls to evaluate their effectiveness in reducing 30-day readmissions post-discharge in neurosurgery patients [32]. However, only 45% of the 83 patients received follow-up calls, impacting the reliability of the study's assessment of readmission rates and post-discharge care. Another study by Phil Edwards investigated the challenges of long-term follow-up data collection in non-commercial, academically-led breast cancer clinical trials [33]. Based on the UK perspective, the study highlighted difficulties in patient compliance and data collection in long-term follow-up studies. Low response rates made data collection challenging, affecting the reliability of the study outcomes. The researchers suggested that maintaining continuous contact with patients plays a crucial role in securing long-term follow-up data. Comparing these studies, our research also faced challenges with follow-up calls, which may have limited the study's results. If a higher response rate had been achieved, the diagnostic accuracy might have exceeded 23%. This highlights the difficulty of data collection through follow-up calls and indicates the need for new approaches to address these challenges in future research.

6) Reformulate the conclusion in a way to show the main findings of the study as well as the limitations.  

☞ Thank you for your comment. We tried to show the main findings of the study as well as the limitations.

☞ [Previous] (Line 311-320) We demonstrated the effectiveness of remote cardiovascular monitoring during daily activities using IoMT devices by diagnosing actual heart disease with 23% accuracy through monitoring abnormal ECG signals collected from 2,000 general hikers using the patch-type electrocardiograph, HiCardi. Furthermore, to validate the practical effectiveness of the data collected in this study, we plan to conduct an in-depth kinetic study through the metabolic equivalents (METs) analysis from a kinematic perspective as a further study. This is significant as a more convenient method for monitoring and diagnosing cardiovascular diseases compared to treadmill tests conducted in hospitals. It is also expected to contribute to preventative healthcare by enabling immediate actions such as hospital visits or medication intake.

[Revised] (Line 433-447) We conducted a pilot study on remote monitoring of ECG signals measured by a patch-type portable IoMT device targeting 2,000 ordinary people. In a situation where face-to-face medical treatment by a doctor was the legal principle, it was thought that it has great implications in that it has contributed to preventive health management by detecting cardiovascular diseases in the daily life of ordinary people under conditions where non-face-to-face remote monitoring was temporarily permitted. However, this study was limited in that it emphasized remote monitoring and reported somewhat descriptive analysis results. In future studies, it would be necessary to propose a model that predicts or learns cardiovascular diseases by analyzing ECG signals with artificial intelligence. Furthermore, to validate the practical effectiveness of the data collected in this study, we plan to conduct an in-depth kinetic study through the metabolic equivalents (METs) analysis from a kinematic perspective as a further study. This is significant as a more convenient method for monitoring and diagnosing cardiovascular diseases compared to treadmill tests conducted in hospitals. It is also expected to contribute to preventative healthcare by enabling immediate actions such as hospital visits or medication intake.

7) The number of used references is embarrassing. I advise to add more recent references.

☞ Thank you for pointing out our leakage. We fully understood and tried to add more recent references.

Reviewer 2 Report

Comments and Suggestions for Authors

This paper presents a clinical trial assessing the effectiveness of a remote monitoring system based on the Internet of Medical Things (IoMT) using a portable electrocardiogram (ECG) device called HiCardi. Conducted with 2,000 participants engaging in hiking activities, the study aimed to evaluate the potential of real-time cardiovascular monitoring for early detection of abnormalities. Participants wore the HiCardi device, which tracked ECG, heart rate, respiration, and body temperature. Abnormal signals detected by remote physicians led to notifications for participants to seek medical attention. Out of 318 participants with abnormal signals, 139 responded, and 30 underwent further examination, resulting in the diagnosis of cardiovascular diseases in seven individuals. The study concluded that the portable ECG device using IoMT technology successfully detected abnormal cardiovascular signals, leading to timely diagnoses and interventions, highlighting its potential for broad application in preventative healthcare.

1. The introduction section is far from expectation.

a. The authors should add more background information for cardiovascular diseases, e.g., other approaches other than IoMT.

b. The authors should give a general introduction to the remote monitoring system and the importance of ECG signals.

c. The authors should summarize existing research gaps.

d. The authors should summarize their main contributions in bullets.

2. A separate related work section should be added to give a comprehensive discussion for relevant studies.

3. Based on "2. Materials and Methods" section, the authors only conduct a usage case study in this study, making it not strong to be accepted.

4. "3. Results" section only presents some descriptive analysis for the survey results. The authors should clearly state their research question and answer these questions by citing the relevant tables and statistical results. Otherwise this paper reads like "what we did and just accpet it if possible".

5. In the conclusion, "23% accuracy" is not a strong evidence for "We demonstrated the effectiveness of remote cardiovascular monitoring". More research is expected for this study.

Author Response

※ Comments and Suggestions for Authors

This paper presents a clinical trial assessing the effectiveness of a remote monitoring system based on the Internet of Medical Things (IoMT) using a portable electrocardiogram (ECG) device called HiCardi. Conducted with 2,000 participants engaging in hiking activities, the study aimed to evaluate the potential of real-time cardiovascular monitoring for early detection of abnormalities. Participants wore the HiCardi device, which tracked ECG, heart rate, respiration, and body temperature. Abnormal signals detected by remote physicians led to notifications for participants to seek medical attention. Out of 318 participants with abnormal signals, 139 responded, and 30 underwent further examination, resulting in the diagnosis of cardiovascular diseases in seven individuals. The study concluded that the portable ECG device using IoMT technology successfully detected abnormal cardiovascular signals, leading to timely diagnoses and interventions, highlighting its potential for broad application in preventative healthcare.

  1. The introduction section is far from expectation.
  2. The authors should add more background information for cardiovascular diseases, e.g., other approaches other than IoMT.

☞ Thank you for your comment. We have tried to supplement the Introduction section with the background and importance of conducting this study.

☞ [Revised] (Line 39-121) In South Korea, remote monitoring in healthcare encounters significant legal obstacles due to the Medical Service Act, particularly Article 34, which prohibits remote diagnosis and treatment between doctors and patients [1]. This article mandates that medical services must be delivered face-to-face, emphasizing the necessity of in-person interactions for diagnosis and treatment. Although the first telemedicine pilot project was attempted in 1988, the debate over its introduction continues today. On the one hand, promoting and activating the telemedicine business along with the development of information and communication technology is an important task for the country in terms of industrial entry, and urges the promotion of the project, saying that science and technology will benefit the public by providing quality medical services. On the other hand, they say that medical practice that does not require face-to-face contact is still risky and that there is no need to pursue this project while taking such risks [2]. As a result, the law restricts the growth of telemedicine and remote monitoring technologies within the country.

Conversely, countries like the United States and those in the European Union have adopted more liberal legal frameworks and economic incentives, which facilitate the rapid development and widespread integration of remote monitoring technologies into their healthcare systems. These nations have embraced telemedicine and remote monitoring, recognizing their potential to enhance healthcare access, improve patient outcomes, and reduce costs. In the United States, remote monitoring has been extensively utilized to manage various health conditions, including COVID-19, chronic obstructive pulmonary disease (COPD), and heart failure. In Japan, remote monitoring is also being applied in diverse healthcare settings, including neonatal care and immunization safety monitoring. In the United States, the Providence health system implemented a Home Monitoring Program (HMP) for COVID-19 patients in March 2020. This program utilized at-home pulse oximeters, thermometers, and text-based surveys to monitor symptoms. The HMP was found to be highly effective, with over 80% engagement in daily text-based surveys and high levels of comfort using home monitoring devices among both English and Spanish-speaking participants. Additionally, the program was associated with increased outpatient and emergency department encounters, indicating that patients felt safer and more satisfied monitoring their condition from home [3]. Similarly, remote patient monitoring for chronic diseases such as COPD has shown potential benefits, including early detection of exacerbations, prompt access to therapy, and ultimately improved patient outcomes [4]. In heart failure management, telemonitoring strategies became routine during the pandemic, with evidence supporting the efficacy and safety of virtual visits and remote vital sign monitoring. The intensity of monitoring was tailored to match the patient's risk profile, highlighting the importance of a structured approach to telehealth implementation [5]. In Japan, remote monitoring is utilized in various healthcare contexts, including the monitoring of adverse events following immunization (AEFI). The Japanese AEFI monitoring system ensures the safe implementation of immunizations by tracking and addressing adverse events. This system is complemented by a unique compensation mechanism, known as the relief system, which provides support for individuals affected by immunization-related issues [6]. Additionally, remote monitoring is employed in neonatal intensive care units (NICUs) in Japan. The use of transcutaneous (tc) measurements of partial pressure of oxygen (tcPO2) and carbon dioxide (tcPCO2) in NICUs allows for continuous and non-invasive monitoring of critically ill neonates, ensuring timely interventions and better health outcomes [7]. As such, Korea has a problem that reduces market dynamism due to positive regulations based on laws that list only those things that are permitted, thereby hindering the growth of the industry along with the flow of the Fourth Industrial Revolution [8].

As a measure to solve this problem, the Ministry of SMEs and Startups of Korea has established a system to designate the Regulation Free Special Zone (RFZ) by region and to ease key regulations related to new businesses as a package, in order to create an environment that can focus on rapidly changing technological industrialization in the era of the 4th Industrial Revolution [9]. First of all, a study analyzing the legal and institutional issues of regulations recognized as obstacles in industrial sites was reported [10-12], and in 2019, the Gangwon Digital Healthcare Regulation Free Special Zone was designated to make it grow the digital healthcare industry, making it possible for Gangwon State to conduct empirical research as a test bed for regulatory improvement of various laws, as shown in Table 1 below.

Table 1. Regulation evidences and substantiation.

Regulation Evidence

Substantiation

MEDICAL SERVICE ACT

-          Diabetes management service based on medical information

-          Blood pressure management service based on medical information

-          Chronic disease from home monitoring service

-          Bio-signal monitoring healthcare service

Article 34 (Remote Medical Treatment)

MEDICAL SERVICE ACT

-          Portable X-ray diagnosis system

Article 37 (Radiation Generating Devices for Diagnostic)

PHARMACEUTICAL AFFAIRS ACT

-          Vaccine demand prediction service (DUR demonstration)

Article 23-3 (Establishment and Operation of Information System for Safe Use of Drugs)

In this study, in order to ease the regulations under Article 34 of the Medical Act of Korea, a bio-signal monitoring healthcare service was planned that partially applied that wearable devices for bio-signal monitoring would be provided and that a doctor in the hospital would be able to monitor electrocardiogram (ECG) for rapid treatment and rescue in emergency situations. Then, an empirical study was conducted on remote monitoring between a doctor and patients depending on the limited scope, location, and method. ECG is a primary diagnostic tool for various cardiovascular diseases, including arrhythmias, myocardial infarctions, and other cardiac abnormalities. The high diagnostic accuracy of ECGs in detecting myocardial ischemia was demonstrated, reinforcing the critical role of ECGs in early detection and treatment planning for cardiovascular conditions [13]. Also, studies such as those by Mincholé and Rodríguez have highlighted the predictive power of ECG signals in identifying patients at risk of future cardiac events, further underscoring the importance of ECG-based diagnostics [14]. In addition, remote monitoring of ECG signals has become increasingly important, especially with the rise of chronic heart disease and the need for continuous, real-time monitoring. A report by the American Heart Association (AHA) in 2019 emphasized that remote monitoring can significantly improve patient outcomes by enabling early detection of potential issues, thereby allowing for timely medical interventions [15]. It was found that remote monitoring of heart failure patients reduced hospitalizations and improved survival rates. These findings highlight the critical role of remote ECG monitoring in managing cardiovascular health, especially in at-risk populations [16].

  1. The authors should give a general introduction to the remote monitoring system and the importance of ECG signals.

☞ Thank you for your good advice. We tried to emphasize the introduction to remote monitoring and the importance of ECG signals.

☞ [Previous] (Line 49-56) Remote monitoring of ECG signals has become increasingly important, especially with the rise of chronic heart disease and the need for continuous, real-time monitoring. A report by the American Heart Association (AHA) in 2019 emphasized that remote monitoring can significantly improve patient outcomes by enabling early detection of potential issues, thereby allowing for timely medical interventions [4]. It was found that remote monitoring of heart failure patients reduced hospitalizations and improved survival rates. These findings highlight the critical role of remote ECG monitoring in managing cardiovascular health, especially in at-risk populations [5].

   [Revised] (Line 122-147) However, despite the various studies reported on ECG signal monitoring, some shortcomings may affect its efficacy and reliability. These shortcomings can be broadly categorized into device power, signal quality, and practical use. One of the primary disadvantages is related to the power constraints of ECG monitoring devices, particularly implantable ones. The limited battery capacity restricts the functionalities of these devices, preventing them from achieving their full potential. Although wireless power transfer (WPT) technology has been proposed to address this issue, the implementation and efficiency of such systems in real-world scenarios remain to be thoroughly validated [17]. Signal quality is another significant challenge in ECG monitoring. During data acquisition and transmission, various artifacts can degrade the ECG signal, leading to incorrect clinical diagnoses. These artifacts may stem from environmental noise, patient movement, or device-related issues. While advanced signal processing techniques such as wavelet transforms, neural networks, and adaptive filters have been developed to suppress these artifacts, ensuring consistently high-quality signals remains a complex task. The proposed multistage adaptive filter aims to tackle multiple types of artifacts, but it requires prior knowledge of the interferences, which may not always be feasible [18]. Wearable ECG devices, such as patch-type monitors, introduce their own set of challenges. These devices, while unobtrusive and user-friendly, are typically suited for short to medium-term monitoring (days to weeks) rather than long-term continuous use. Moreover, achieving a balance between low power consumption and high diagnostic accuracy in these devices remains a significant challenge. Existing schemes often suffer from high power consumption to maintain accuracy, which can limit their practicality [19]. Another critical issue is the difficulty in ensuring the clinical acceptability of ECG signals in unsupervised environments. Real-time signal quality assessment (SQA) methods have been proposed to classify the acquired ECG signal into acceptable or unacceptable categories, but these methods need to be robust and efficient to be effective in practical applications [20].

  1. The authors should summarize existing research gaps. The authors should summarize their main contributions in bullets.

☞ Thank you for your good comments We tried to add similar studies along with the shortcomings of the patch-type ECG device and remote monitoring in the Introduction section.

☞ [Revised] (Line 107-147) ECG is a primary diagnostic tool for various cardiovascular diseases, including arrhythmias, myocardial infarctions, and other cardiac abnormalities. The high diagnostic accuracy of ECGs in detecting myocardial ischemia was demonstrated, reinforcing the critical role of ECGs in early detection and treatment planning for cardiovascular conditions [13]. Also, studies such as those by Mincholé and Rodríguez have highlighted the predictive power of ECG signals in identifying patients at risk of future cardiac events, further underscoring the importance of ECG-based diagnostics [14]. In addition, remote monitoring of ECG signals has become increasingly important, especially with the rise of chronic heart disease and the need for continuous, real-time monitoring. A report by the American Heart Association (AHA) in 2019 emphasized that remote monitoring can significantly improve patient outcomes by enabling early detection of potential issues, thereby allowing for timely medical interventions [15]. It was found that remote monitoring of heart failure patients reduced hospitalizations and improved survival rates. These findings highlight the critical role of remote ECG monitoring in managing cardiovascular health, especially in at-risk populations [16].

However, despite the various studies reported on ECG signal monitoring, some shortcomings may affect its efficacy and reliability. These shortcomings can be broadly categorized into device power, signal quality, and practical use. One of the primary disadvantages is related to the power constraints of ECG monitoring devices, particularly implantable ones. The limited battery capacity restricts the functionalities of these devices, preventing them from achieving their full potential. Although wireless power transfer (WPT) technology has been proposed to address this issue, the implementation and efficiency of such systems in real-world scenarios remain to be thoroughly validated [17]. Signal quality is another significant challenge in ECG monitoring. During data acquisition and transmission, various artifacts can degrade the ECG signal, leading to incorrect clinical diagnoses. These artifacts may stem from environmental noise, patient movement, or device-related issues. While advanced signal processing techniques such as wavelet transforms, neural networks, and adaptive filters have been developed to suppress these artifacts, ensuring consistently high-quality signals remains a complex task. The proposed multistage adaptive filter aims to tackle multiple types of artifacts, but it requires prior knowledge of the interferences, which may not always be feasible [18]. Wearable ECG devices, such as patch-type monitors, introduce their own set of challenges. These devices, while unobtrusive and user-friendly, are typically suited for short to medium-term monitoring (days to weeks) rather than long-term continuous use. Moreover, achieving a balance between low power consumption and high diagnostic accuracy in these devices remains a significant challenge. Existing schemes often suffer from high power consumption to maintain accuracy, which can limit their practicality [19]. Another critical issue is the difficulty in ensuring the clinical acceptability of ECG signals in unsupervised environments. Real-time signal quality assessment (SQA) methods have been proposed to classify the acquired ECG signal into acceptable or unacceptable categories, but these methods need to be robust and efficient to be effective in practical applications [20].

  1. A separate related work section should be added to give a comprehensive discussion for relevant studies.

☞ Thank you for your sincere comment. We tried to give a comprehensive discussion for relevant studies.

☞ [Previous] (Line 243-309) This study conducted a clinical trial on 2,000 ordinary hikers to verify the effectiveness of the remote monitoring technology using the IoMT devices and system. As a result, 7 of 2,000 hikers in real-time using the IoMT devices and system were real-diagnosed with heart diseases. In the case of some heart disease diagnoses, many cases are difficult to detect in the existing monitoring system, and monitoring for more than a week is required in some cases. In this study, it was meaningful to test that the remote IoMT monitoring system and device in daily activities can be effective in detecting heart disease. Similarly, in the study by Paddy M. Barrett et al [9], comparing 24-hour Holter monitors with 14-day adhesive patch monitors in 146 patients, the patch monitor detected more arrhythmic events. Furthermore, in the study by Warren M. Smith et al [10], using a P-wave-centric ECG patch monitor, 46% of the 50 patients detected rhythm changes requiring management alterations, compared to 12% with the Holter monitor. These studies suggest that the IoMT device may offer higher diagnostic rates and convenience compared to traditional Holter monitors.

In this study, approximately 15 types of heart disease-related abnormal signals were observed using the IoMT device, including PVCs and atrial fibrillation. Among these, PVCs were the most frequently observed, with 79 instances. Examining other studies related to remote ECG monitoring, Markus Lueken et al [11] focused on the automated signal quality assessment of single-lead ECG recordings for the early detection of various arrhythmias, including PVCs. This study reported that PVCs were observed in approximately 75% of participants during 24 to 48-hour ambulatory ECG monitoring, demonstrating a high prevalence of these abnormal signals in the monitored population. Similarly, in the study by B. Venkataramanaiah et al [12], the prevalence and characteristics of premature ventricular contractions (PVCs) were examined using remote ECG monitoring systems. This study emphasized the importance of detecting PVCs due to their potential to develop into more serious arrhythmias. The results indicated that PVCs were observed in 67.7% of the monitored population via 24-hour Holter monitoring. Among these, 60.0% had a PVC burden of less than 5%, while 7.7% had a PVC burden of 5% or more. As shown in our study and others, the occurrence rate of PVCs among abnormal ECG signals is relatively high in both IoMT devices and Holter monitoring systems. This is because PVCs can be precursors to various serious arrhythmias [13].

Additionally, our study found that 7 out of 30 patients (23%) who visited the hospital were diagnosed with actual heart disease using the IoMT device. This indicates that remote monitoring can have clinical effectiveness in preventative healthcare. A similar study by Marco V. Perez et al [14] involved 419,297 participants wearing Apple Watches, where those with detected irregular pulses acquised ECG patches for 7 days of monitoring. Among the 2,161 participants who received irregular pulse alarms, 450 returned ECG patches, and 34% of these showed atrial fibrillation. The positive predictive value of the irregular pulse alarms was 84%. The Fitbit Heart Study by Steven A. Lubitz et al [15] followed a similar approach, analyzing 455,699 participants. Those with detected irregular heart rhythms were provided with an ECG patch for one week. The Fitbit PPG-based algorithm detected irregular heart rhythms and used ECG patches to confirm atrial fibrillation. Results indicated a high positive predictive value of 98%, demonstrating the effectiveness of consumer wearable technology in identifying AF episodes. Compared to these two studies, our study's diagnostic accuracy appears relatively lower. However, this could be attributed to factors such as sweat, heavy breathing, or movement during hiking causing poor contact with the patch, and geographic characteristics of being in the mountains potentially affecting the transmission of ECG signals. While our study's accuracy of diagnosis is relatively lower, this can be attributed to the limitation of a small number of confirmed follow-up calls.

Moreover, a small number of confirmed follow-up calls is a limitation of our study. Many other studies also face low compliance with follow-up calls. For example, in a study by Dorothy M. Mwachiro et al [16], nurses conducted follow-up calls to evaluate their effectiveness in reducing 30-day readmissions post-discharge in neurosurgery patients. However, only 45% of the 83 patients received follow-up calls, impacting the reliability of the study's assessment of readmission rates and post-discharge care. Another study by Phil Edwards [17] investigated the challenges of long-term follow-up data collection in non-commercial, academically-led breast cancer clinical trials. Based on the UK perspective, the study highlighted difficulties in patient compliance and data collection in long-term follow-up studies. Low response rates made data collection challenging, affecting the reliability of the study outcomes. The researchers suggested that maintaining continuous contact with patients plays a crucial role in securing long-term follow-up data. Comparing these studies, our research also faced challenges with follow-up calls, which may have limited the study's results. If a higher response rate had been achieved, the diagnostic accuracy might have exceeded 23%. This highlights the difficulty of data collection through follow-up calls and indicates the need for new approaches to address these challenges in future research.

[Revised] (Line 354-431) This study was an empirical research that remotely monitored single-lead ECG signals using a patch-type portable IoMT device. The device detected abnormal signals of atrial fibrillation (AF), arterial arrhythmia (SA), premature atrial contractions (PAC), bundle branch block (BBB), atrioventricular (AV) block, premature ventricular contractions (PVC), ventricular fibrillation (VF), tachycardia, bradycardia, ST depression, chest pain, palpitations, and other abnormal rhythms. In this study, it was meaningful to test that the remote IoMT monitoring system and device in daily activities can be effective in detecting cardiovascular disease. Paddy M. Barrett et al. compared 24-hour Holter monitors with 14-day adhesive patch monitors in 146 patients, the patch monitor detected more arrhythmic events [25]. Furthermore, in the study by Warren M. Smith et al., using a P-wave-centric ECG patch monitor, 46% of the 50 patients detected rhythm changes requiring management alterations, compared to 12% with the Holter monitor [26]. These studies suggest that the IoMT device may offer higher diagnostic rates and convenience compared to traditional Holter monitors. In addition, Choi et al. focused on developing a low-cost, non-invasive heart monitoring system for early AF detection [17]. The device analyzed heartbeat variations and RR intervals in the ECG signal, generating alerts when abnormal patterns indicative of AF were detected. This real-time monitoring system was designed for home use, providing a practical solution for individuals with AF concerns. Similar to these previous studies, the detection of abnormal signals related to cardiovascular diseases using single-lead ECG portable devices has been extensively studied, showcasing the potential of these devices in everyday healthcare monitoring.

In this study, approximately 15 types of cardiovascular disease-related abnormal signals were observed using the IoMT device, including AF, SA, PACs, BBB, AV block, PVCs, VF, tachycardia, bradycardia, ST depression, chest pain, and palpitations. Among these, PVCs were the most frequently observed, with 79 instances. Examining other studies related to remote ECG monitoring, Markus Lueken et al. focused on the automated signal quality assessment of single-lead ECG recordings for the early detection of various arrhythmias, including PVCs [27]. This study reported that PVCs were observed in approximately 75% of participants during 24 to 48-hour ambulatory ECG monitoring, demonstrating a high prevalence of these abnormal signals in the monitored population. Similarly, in the study by B. Venkataramanaiah et al [28], the prevalence and characteristics of premature ventricular contractions (PVCs) were examined using remote ECG monitoring systems. This study emphasized the importance of detecting PVCs due to their potential to develop into more serious arrhythmias. The results indicated that PVCs were observed in 67.7% of the monitored population via 24-hour Holter monitoring. Among these, 60.0% had a PVC burden of less than 5%, while 7.7% had a PVC burden of 5% or more. As shown in our study and others, the occurrence rate of PVCs among abnormal ECG signals is relatively high in both IoMT devices and Holter monitoring systems. This is because PVCs can be precursors to various serious arrhythmias [29].

Additionally, our study found that 7 out of 30 patients (23%) who visited the hospital were diagnosed with actual heart disease using the IoMT device. This indicates that remote monitoring can have clinical effectiveness in preventative healthcare. A similar study by Marco V. Perez et al. involved 419,297 participants wearing Apple Watches, where those with detected irregular pulses acquired ECG patches for 7 days of monitoring [30]. Among the 2,161 participants who received irregular pulse alarms, 450 returned ECG patches, and 34% of these showed atrial fibrillation. The positive predictive value of the irregular pulse alarms was 84%. The Fitbit Heart Study by Steven A. Lubitz et al. followed a similar approach, analyzing 455,699 participants [31]. Those with detected irregular heart rhythms were provided with an ECG patch for one week. The Fitbit PPG-based algorithm detected irregular heart rhythms and used ECG patches to confirm atrial fibrillation. Results indicated a high positive predictive value of 98%, demonstrating the effectiveness of consumer wearable technology in identifying AF episodes. Similar to these two studies, remote monitoring could be expected to have clinical effects on preventive health management in this study. However, it had the limitation that the sample size of the empirical research was small, making it difficult to have representativeness. In addition, since the empirical research was conducted in a designated location, factors such as sweat, shortness of breath, or movement during the hike, which may have caused poor contact with the patch, and the geographical characteristics of the mountain potentially affected the transmission of the ECG signal, which may have affected the results.

Lastly, a limitation of this study is that the compliance rate among those who responded to the recommendations for hospital treatment after discovering abnormal signs suggestive of cardiovascular disease and actually visiting the hospital was low. Similarly, many other studies also face low compliance with follow-up calls. For example, in a study by Dorothy M. Mwachiro et al., nurses conducted follow-up calls to evaluate their effectiveness in reducing 30-day readmissions post-discharge in neurosurgery patients [32]. However, only 45% of the 83 patients received follow-up calls, impacting the reliability of the study's assessment of readmission rates and post-discharge care. Another study by Phil Edwards investigated the challenges of long-term follow-up data collection in non-commercial, academically-led breast cancer clinical trials [33]. Based on the UK perspective, the study highlighted difficulties in patient compliance and data collection in long-term follow-up studies. Low response rates made data collection challenging, affecting the reliability of the study outcomes. The researchers suggested that maintaining continuous contact with patients plays a crucial role in securing long-term follow-up data. Comparing these studies, our research also faced challenges with follow-up calls, which may have limited the study's results. If a higher response rate had been achieved, the diagnostic accuracy might have exceeded 23%. This highlights the difficulty of data collection through follow-up calls and indicates the need for new approaches to address these challenges in future research.

  1. Based on "2. Materials and Methods" section, the authors only conduct a usage case study in this study, making it not strong to be accepted.

☞ Thank you for pointing out the weakness of the section. We tried to make it strong to be accepted. In details, we added the several sentences about inclusion and exclusion criteria, and specification of ECG monitoring device.

☞ [Previous] (Line 75-93)

2.1. Participants

In this study, a clinical trial was conducted to verify the effectiveness of the remote monitoring diagnosis technology using the IoMT device. The clinical trial was conducted a total of 30 times in about 8 months and a total of 2,000 hikers participated. Each hiker filled out a survey about the clinical trial consent and existing health information in advance then attached a wearable ECG monitoring patch to their chest and climbed the mountain. This study was carried out with approval from the Research Ethics Committee of Yonsei University Wonju Severance Christian Hospital (IRB approval number: CR319186) and registered with the Clinical Research Information Service (CRIS) operated by the Korea Disease Control and Prevention Agency (CRIS trial registration number: KCT0005795).

2.2. ECG monitoring device

HiCardi® (MEZOO Co. Ltd., Wonju, Gangwon State, Republic of Korea), an IoMT device used in this study, is a patch-type electrocardiograph attached to the hiker's chest. This wearable device could monitor and record various parameters related to heart signals such as a single-lead ECG, heart rate, respiration, skin surface temperature, and activity. The data from the wearable patch were transferred through Bluetooth low energy to a mobile gateway, which was implemented as a smartphone application. The mobile gateway transmitted the real-time data to the core, analytics, and service platform [8].

   [Revised] (Line 171-205)

2.1. Participants

In this study, a clinical trial was conducted to verify the effectiveness of the remote monitoring diagnosis technology using the IoMT device. The clinical trial was conducted a total of 30 times in about 8 months and a total of 2,000 hikers participated. The inclusion criteria were adults aged 19 years or older, who gave written consent to participate in the empirical research, and who could understand and follow the on-site training and instructions. Also, the exclusion criteria were those who could not attach the patch-type IoMT device due to skin disease, those who were pregnant, those who were participating in another clinical trial at the same time, or those who had recently completed participation in a previous clinical trial. Each hiker filled out a survey about the clinical trial consent and existing health information in advance then attached a wearable ECG monitoring patch to their chest and climbed the mountain. This study was carried out with approval from the Research Ethics Committee of Yonsei University Wonju Severance Christian Hospital (IRB approval number: CR319186) and registered with the Clinical Research Information Service (CRIS) operated by the Korea Disease Control and Prevention Agency (CRIS trial registration number: KCT0005795).

2.2. ECG monitoring device

HiCardi® (MEZOO Co. Ltd., Wonju, Gangwon State, Republic of Korea), an IoMT device used in this study, is a patch-type electrocardiograph. This wearable device could monitor and record various parameters related to heart signals such as a single-lead ECG, heart rate, respiration, skin surface temperature, and activity. It has a comprehensive set of specifications aimed at providing accurate health monitoring data. The heart rate (HR) range of the device is from 0 to 300 beats per minute (bpm), with a precision of ±2 bpm or ±2%, whichever is greater. The temperature monitoring range is from 32ËšC to 43ËšC, with an accuracy of ±0.3%. It also provides respiration rate measurements ranging from 5 to 60 breaths per minute or up to 120 breaths per minute, maintaining an accuracy of ±2 bpm or ±2%. The device offers an impressive battery operation time of 72 to 168 hours with a charging time of 1 to 2 hours. It also features water protection capabilities from IPX2 to IP67, ensuring durability and reliability in various conditions. The memory capacity of the device supports data recording for up to 7 days, with optional expansions available. Despite its robust functionality, the device maintains a lightweight design, weighing less than 10 grams (0.35 oz). The data from the wearable patch were transferred through Bluetooth low energy to a mobile gateway, which was implemented as a smartphone application. The mobile gateway transmitted the real-time data to the core, analytics, and service platform [24].

  1. "3. Results" section only presents some descriptive analysis for the survey results. The authors should clearly state their research question and answer these questions by citing the relevant tables and statistical results. Otherwise this paper reads like "what we did and just accpet it if possible".

☞ Thank you for your sincere feedback. We fully understood and tried to state our research question and answer clearly. First, in the Introduction section, we described in detail the background for conducting this study, and second, in the Conclusion section, we tried to clearly state the limitations of this study and topics that could be conducted in future studies.

  1. In the conclusion, "23% accuracy" is not a strong evidence for "We demonstrated the effectiveness of remote cardiovascular monitoring". More research is expected for this study.

☞ Thank you for your opinion. We fully understood that it was not a strong evidence. Also, we tried to revise the conclusion to make it more logical.

☞ [Previous] (Line 311-320) We demonstrated the effectiveness of remote cardiovascular monitoring during daily activities using IoMT devices by diagnosing actual heart disease with 23% accuracy through monitoring abnormal ECG signals collected from 2,000 general hikers using the patch-type electrocardiograph, HiCardi. Furthermore, to validate the practical effectiveness of the data collected in this study, we plan to conduct an in-depth kinetic study through the metabolic equivalents (METs) analysis from a kinematic perspective as a further study. This is significant as a more convenient method for monitoring and diagnosing cardiovascular diseases compared to treadmill tests conducted in hospitals. It is also expected to contribute to preventative healthcare by enabling immediate actions such as hospital visits or medication intake.

   [Revised] (Line 433-447) We conducted a pilot study on remote monitoring of ECG signals measured by a patch-type portable IoMT device targeting 2,000 ordinary people. In a situation where face-to-face medical treatment by a doctor was the legal principle, it was thought that it has great implications in that it has contributed to preventive health management by detecting cardiovascular diseases in the daily life of ordinary people under conditions where non-face-to-face remote monitoring was temporarily permitted. However, this study was limited in that it emphasized remote monitoring and reported somewhat descriptive analysis results. In future studies, it would be necessary to propose a model that predicts or learns cardiovascular diseases by analyzing ECG signals with artificial intelligence. Furthermore, to validate the practical effectiveness of the data collected in this study, we plan to conduct an in-depth kinetic study through the metabolic equivalents (METs) analysis from a kinematic perspective as a further study. This is significant as a more convenient method for monitoring and diagnosing cardiovascular diseases compared to treadmill tests conducted in hospitals. It is also expected to contribute to preventative healthcare by enabling immediate actions such as hospital visits or medication intake.

Round 2

Reviewer 1 Report

Comments and Suggestions for Authors

The authors answered all my comments and concerns. I suggest the publication of the paper.

Reviewer 2 Report

Comments and Suggestions for Authors

No further comments